# Positioning of Unmanned Underwater Vehicle Based on Autonomous Tracking Buoy

**DOI:** 10.3390/s23094398

**Published:** 2023-04-29

**Authors:** Yuhan Li, Ruizhi Ruan, Zupeng Zhou, Anqing Sun, Xiaonan Luo

**Affiliations:** 1School of Mechanical and Electrical Engineering, Guilin University of Electronic Technology, Guilin 541004, China; liyuhan038@guet.edu.cn; 2School of Information and Communication, Guilin University of Electronic Technology, Guilin 541004, China; 3School of Computer and Information Security, Guilin University of Electronic Technology, Guilin 541004, China

**Keywords:** buoy, unmanned underwater vehicle, dynamic tracking, ultra-short baseline matrix

## Abstract

This paper presents a novel method for the dynamic positioning of an unmanned underwater vehicle (UUV) with unknown trajectories based on an autonomous tracking buoy (PUVV-ATB) that indirectly positions the UUV using ultra-short baseline measurements. The method employs a spatial location geometric model and divides the positioning process into four steps, including data preprocessing to detect geometric errors and apply mean filtering, direction capture, position tracking, and position synchronization. To achieve these steps, a new adaptive tracking control algorithm is proposed that does not require trajectory prediction and is applied to the last three steps. The algorithm is deployed to the buoy for tracking simulation and sea trial experiments, and the results are compared with those of a model predictive control algorithm. The autonomous tracking buoy based on the adaptive tracking control algorithm runs more stably and can better complete the precise tracking task for the UUV with a positioning error of less than 10 cm. This method breaks the premise of trajectory prediction based on traditional tracking control algorithms, providing a new direction for further research on UUV localization. Furthermore, the conclusion of this paper has important reference value for other research and application fields related to UUV.

## 1. Introduction

With the continued development of the marine industry, there has been a rise in human exploration and utilization of marine resources. To ensure safety and improve the efficiency of underwater operations, unmanned underwater vehicles (UUVs) are being increasingly utilized, gradually replacing professional divers [1]. However, the high-precision and large-scale positioning of UUVs remains a challenge that must be addressed due to the complex nature of the marine environment. The successful adoption and application of UUVs depend on solving this challenge [2].

### 1.1. Review of Previous Work

Dynamic target positioning methods commonly used include positioning through an inertial navigation IMU, vision-based positioning, underwater sonar-based positioning, multi-sensor fusion-based positioning, buoy-assisted positioning, and others. Of these methods, inertial navigation and IMU positioning are susceptible to underwater ocean currents, leading to significant positioning errors [3]. In addition, these errors accumulate over time, leading to reduced accuracy. Meanwhile, the accuracy of vision-based positioning is influenced by the clarity and color of the water [4]. The accuracy and range of underwater positioning are severely limited due to low underwater visibility and interference from underwater microorganisms. So underwater sonar positioning is commonly used to determine starting positions, and the positioning of moving targets requires sensor fusion with other sensors [5,6]. Multi-sensor fusion positioning is currently one of the mainstream dynamic target positioning methods, and it is more effective in relatively clear waters with small waves [7,8,9]. However, its positioning ability is extremely limited in an ocean environment with strong waves, ocean currents, and low visibility. Buoy-assisted positioning is the current mainstream underwater positioning method [10]. It uses a positioning system composed of multiple sonars to locate underwater targets. This method is less affected by seawater clarity and ocean currents and can provide stable and reliable positioning for underwater targets. However, most of the buoys used for positioning assistance are stationary. Due to the limited detection range of the sonar sensor and its sensitivity to distance, the buoy is unable to effectively locate targets that exceed the detection range of underwater sonar. Therefore, this method is not suitable for positioning UUVs operating in large areas.

Most of the existing research in the field of UUV focuses on underwater trajectory planning and tracking, while the core issue of tracking control of dynamic targets without trajectory prediction is relatively unexplored. For instance, Cao et al., proposed an autonomous UUV tracking control method based on trajectory prediction, using deep learning to recognize underwater targets with the multibeam forward-looking sonar image and constructing a time-profit Elman neural network to predict the target’s trajectory [11]. However, this method requires a high response speed from the controller, and trajectory prediction errors are inevitable, although it increases the accuracy of dynamic target prediction. Similarly, to ensure the finite-time convergence of motion tracking error and the stability of the entire closed-loop control system, Liu et al. developed a trajectory tracking control scheme for underwater vehicles based on nonlinear disturbance observer back-calculation finite-time sliding mode to ensure the finite-time convergence of motion tracking error and the stability of the entire closed-loop control system [12]. However, this scheme also has high requirements for real-time and precision trajectory prediction. Hu et al. designed a trajectory re-planning controller based on model predictive control (MPC) [13]. The obstacle avoidance function was set in the design of the MPC trajectory re-planning controller so that the re-planning trajectory generated by the re-planning controller can avoid obstacles, greatly optimizing the autonomous underwater vehicles (AUV) trajectory planning and tracking performance. But it is difficult to re-plan the trajectory without the precise position of the AUV. Furthermore, Shen et al. proposed a novel Lyapunov-based model predictive control (LMPC) framework for the AUV to improve the trajectory tracking performance, which improved the robustness of the tracking control [14]. However, it is proposed based on the scheme of MPC and has a gap in trajectory prediction. All methods mentioned above focus solely on underwater trajectory planning or tracking, neglecting the important premise of positioning the UUV. Since the trajectory of the UUV is unknowable before positioning, dynamic target tracking control algorithms that rely on target trajectory points are not suitable for tracking UUVs. Therefore, there is a need for research on UUV positioning to enable tracking without trajectory prediction. 

### 1.2. Novelty and Contributions of the Article

To address the aforementioned limitations, this paper proposes a novel indirect positioning method based on autonomous tracking buoys and presents an adaptive tracking control algorithm without trajectory prediction (ATC-NTP). The dynamic buoy-assisted positioning method has several advantages, including stability, reliability, and high precision, and can overcome the limitations of the limited positioning area. Compared to MPC, the ATC-NTP algorithm does not require high-precision sensors or powerful airborne computers for position prediction and provides better tracking performance. By combining the dynamic buoy with the ATC-NTP algorithm, this method solves the problem of positioning for UUVs. The proposed method provides a new direction for further research on unmanned underwater vehicle localization and has important reference value for other research and application fields in UUV.

## 2. Spatial Location Geometric Model

The geometric model is established in Figure 1 based on the spatial position relationship between the buoy and the UUV. The buoy model is designed using the software SolidWorks (version 2020), and the UUV is created by our laboratory team. Point O represents the center position of the surface buoy, while points A, B, and C indicate the positions of the three hydrophones on the buoy. Point D’ denotes the position of the center acoustic beacon of the UUV, and point D represents the vertical projection of the UUV. The x-axis, y-axis, and z-axis represent the right roll direction, forward direction, and vertical direction of the buoy, respectively. *R* represents the horizontal distance between the center of the buoy and the listener, H denotes the depth of the UUV, $\rho_{1}$, $\rho_{2}$ and $\rho_{3}$ represent the straight-line distances between the three hydrophones A, B, C on the buoy and the acoustic beacon D’ on the UUV. Additionally, $R_{1}$, $R_{2}$ and $R_{3}$ are the vertical projections of $\rho_{1}$, $\rho_{2}$ and $\rho_{3}$ on the horizontal plane, respectively. The line segment OA=OB=OC=*R*, and the angle between OA, OB, and OC is 120 degrees. *R* is a fixed constant; H is measured by the depth gauge on the UUV; $\rho_{1}$, $\rho_{2}$ and $\rho_{3}$ are measured by the USBL [15,16].

According to the spatial position model and geometric theorem in Figure 1, it can be deduced that if $\rho_{1}$ = max {$\rho_{1}$, $\rho_{2}$*,*
$\rho_{3}$}, then point D falls within the range of the opening angle of ∠BOC. Similarly, if $\rho_{2}$ = max {$\rho_{1}$, $\rho_{2}$, $\rho_{3}$}, then point D falls within the opening angle range of ∠AOC. Finally, if $\rho_{3}$ = max {$\rho_{1}$, $\rho_{2}$, $\rho_{3}$}, then point D falls within the opening angle range of ∠AOB. This conclusion provides a theoretical basis for the buoy to capture the horizontal azimuth of the UUV and will be applied in the design of the system.

## 3. System Design of Autonomous Tracking Buoy

The distance $\rho_{1}$, $\rho_{2}$, $\rho_{3}$ are measured using the ultra-short baseline (USBL), Which is a type of underwater acoustic positioning system used to determine the location of subsea objects. In this paper, the USBL system consists of two main components: three hydrophones mounted on the buoy and an acoustic beacon attached to the UUV. The acoustic beacon sends an acoustic signal, which is received by the hydrophones. By measuring the time taken for the signal to travel between the acoustic beacon and the hydrophones, the distance $\rho_{1}$, $\rho_{2}$, $\rho_{3}$ can be determined with high accuracy. The USBL system is highly accurate, with location accuracy typically within a few centimeters. 

The usage mechanism of the USBL system is as follows: First, the clocks of the buoy and the UUV system are synchronized based on the clock of the shore-based operating system. Secondly, the acoustic beacon on the UUV emits a sound wave with a frequency of 10 KHz every 200 ms. The distance between the three hydrophones on the buoy, which is set to $\sqrt{3}R$, is four times the wavelength of the sound wave, which is 60 cm. According to the Shannon sampling theorem, the sound wave sampling rate of the buoy system is set to 40 KHz [17]. The transmission time of the sound wave in the water is equal to the difference between the time corresponding to the peak of the 10 KHz sound wave and the sound wave emission time of the acoustic beacon. The distance $\rho_{1}$, $\rho_{2}$, $\rho_{3}$ can be obtained by multiplying the time difference by the propagation speed of the sound wave. 

After measuring the depth H with the depth gauge, the values of $\rho_{1}$, $\rho_{2}$, $\rho_{3}$ and H are checked to confirm that they satisfy the theorem that the difference between the two sides of a triangle must be smaller than the third side. If any of the values are wrong, the data package will be discarded; otherwise, they will be temporarily stored in the data buffer. Once the data package has been collected ten times, it is averaged to reduce measurement error. According to the collected distance data and the conclusions above, the rough azimuth angle of the UUV relative to the buoy can be determined, and then the yaw angle of the buoy can be adjusted so that the front direction of the buoy points to the water surface projection of the UUV. Once the UUV’s orientation is locked, the buoy starts tracking the UUV. The distance difference between the buoy and the UUV in the horizontal direction is calculated based on the measured data $\rho_{1}$ and *H*, which is used as input for the PID controller to output the expected speed of the buoy. After a period of time, the tracking is successful, and the distance difference is kept within the set error range *δ*. When tracking is successful, the buoy is set to the synchronization state. The position of the UUV is analyzed and synchronized by combining the synchronization error, longitude, and latitude coordinates of the buoy. The operating flow of the system is shown in Figure 2, which is designed according to the process of positioning the UUV with the software draw.io (version 15.8.4.0). 

### 3.1. Data Preprocession

To achieve accurate tracking, it is necessary to apply a data filter due to the inevitability of errors in a small part of the data resulting from various complex interferences in the underwater sonar ranging process. Firstly, the values of $\rho_{1}$, $\rho_{2}$ and $\rho_{3}$ need to meet inequality (1), inequality (2), and inequality (1) according to the geometric theorem mentioned in Section 2.
(1)$${|\rho }_{2}-\rho_{1}|<\sqrt{3}R$$
(2)$${|\rho }_{1}-\rho_{3}|<\sqrt{3}R$$
(3)$${|\rho }_{3}-\rho_{2}|<\sqrt{3}R$$
where $\sqrt{3}R$ is the distance between any two hydrophones on the buoy and is set to 4 times the wavelength of the sound wave, which is equivalent to 60 cm. Accurate values of $\rho_{1}$, $\rho_{2}$ and $\rho_{3}$ are stored in the temporary data register. After accumulating the values of $\rho_{1}$, $\rho_{2}$ and $\rho_{3}$ for ten times, a mean filter is applied using Equation (4) to reduce measurement errors, with *N* taken as 10. In this equation, $\bar{\rho_{i}}$ represents the mean distance between the *i*-th hydrophone and the UUV, while $\rho_{ij}$ is the *j*-th measurement of the distance between the *i*-th hydrophone and the UUV.
(4)$$\bar{\rho_{i}}=\frac{\sum_{j=1}^{N}\rho_{ij}}{N}\;i=1,2,3$$

### 3.2. Capturing UUV

The buoy is equipped with left and right double thrusters, enabling it to adjust its pitch and yaw angles. The orientation of the UUV can be determined according to the data $\rho_{1}$, $\rho_{2}$ and $\rho_{3}$. If point D falls within the opening angle range of ∠AOC, the buoy will spin clockwise. Conversely, if point D is within the opening angle range of ∠AOB, the buoy will spin counterclockwise. If point D is within the opening angle range of ∠BOC, the buoy will spin counterclockwise when $\rho_{2}$ > $\rho_{3}$ and the buoy will spin clockwise when $\rho_{2}$ < $\rho_{3}$**.**
(5)$$\left|\rho_{2}-\rho_{3}\right|\leq \left(1-\sigma \right)\times \sqrt{3}R$$
(6)$$\theta_{max}=\arcsin\frac{\left(1-\sigma \right)\times {(\rho }_{2}+\rho_{3})}{\sqrt{3}R\times \sqrt{2\rho_{2}^{2}+2\rho_{3}^{2}-3R^{2}}}$$

During the buoy spinning process, the system evaluates whether the difference between $\rho_{2}$ and $\rho_{3}$ is small enough by using inequality (5). If $\rho_{2}$ and $\rho_{3}$ meet the inequality (5), the capture is considered successful. Otherwise, the buoy will continue to adjust its yaw until inequality (5) is met. The parameter *σ* is the preset capture accuracy of the system, and its value is within the interval [0, 1]. The time taken from the start of capture to the end is known as the capture time. The UUV’s horizontal azimuth relative to the buoy when capturing successfully is the capture error θ, which is related to σ as shown by Equation (6). The maximum value of θ is $\theta_{max}$, which can be calculated using Equation (6). Once the capture accuracy is set, the capture error is always less than $\theta_{max}$. The larger the σ value, the higher the capture accuracy and the longer the capture time, resulting in a smaller capture error. If the value *σ* is 1, then the value of $\theta_{max}$ is 0°. This capture state is ideal but difficult to achieve in practice. Conversely, the smaller the *σ* value, the lower the capture accuracy and the shorter the capture time, resulting in a larger capture error. It is not advisable to set the value σ too small, as it would render the capture meaningless.

### 3.3. Tracking UUV

When the target is captured, it means that its direction has been locked. Within the allowable range of capture error, it can be considered that the vertical projection of the UUV onto the water surface is directly in front of the buoy’s frond. The buoy then enters the state of tracking the UUV.
(7)$$\sqrt{{|\rho }_{1}^{2}-H^{2}|}<\delta$$

Inequality (7) involves the tracking allowable error parameter δ, which is a preset parameter indicating the allowable horizontal distance difference between the buoy and the UUV when tracking is successful. A smaller value of *δ* results in a longer tracking time and a smaller horizontal position deviation between the buoy and the UUV upon successful tracking. On the contrary, a larger value of δ results in a shorter tracking time and a greater deviation between the buoy and the horizontal position of the UUV upon successful tracking. To evaluate the initial tracking success, inequality (7) is checked at the beginning of tracking, and tracking continues until inequality (7) is satisfied. During tracking, the buoy moves forward, and the speed of forward movement is determined by the difference between $\rho_{1}$ and *H*. Discrete incremental PID control [18] is used for speed adjustment.

At time *t*, the speed increment is calculated using Equations (8) and (9), which take into account the current horizontal distance difference between the buoy and the UUV as well as the accumulated distance difference up to time t.
(8)$$∆P\left(t\right)=K_{p}\left[e\left(t\right)-e\left(t-1\right)\right]+K_{i}e\left(t\right)+K_{d}\left[e\left(t\right)-2e\left(t-1\right)+e\left(t-2\right)\right]$$
(9)$$e\left(n\right)=\sqrt{{|\rho }_{1}^{2}-H^{2}|}\;$$
where ∆*P*(*t*) is the velocity increment from time *t* − 1 to time *t*, *e(t)* is the distance difference between the buoy and the horizontal position of the UUV at time *t*, $K_{p}$, $K_{i}$ and $K_{d}$ are proportional coefficients, integral coefficients, and differential coefficients.

Compute the velocity at the current moment using Equation (10).
(10)$$P\left(t\right)=∆P\left(t\right)+P\left(t-1\right)$$

In Equation (10), $P\left(t\right)$ represents the current velocity, and $P\left(t-1\right)$ represents the velocity at the previous time. The expected speed value calculated is input into the controller, which outputs the corresponding PWM signal to drive the buoy’s motion state change. Throughout this process, the buoy keeps capturing the UUV in parallel to ensure it remains in a tracking state. The algorithm is shown as following Algorithm 1:
**Algorithm 1:** Target Acquisition and Tracking Algorithm.**Input**: The distance between the three hydrophones on the buoy and the central acoustic beacon of the underwater vehicle: *ρ1*, *ρ2*, *ρ3*;    The depth of UUV:*H*;    Capture tolerance and tracking tolerance.**Output**: The speed at which the buoy moves forward at the current moment;
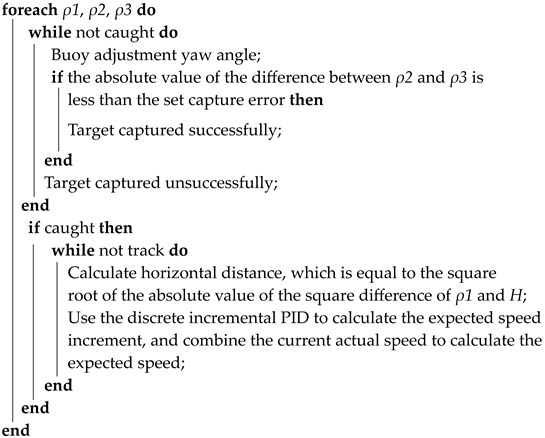
Input the desired speed into the controller to calculate the PWM value of the thruster;

### 3.4. Synchronizing UUV

The buoy system enters the state of synchronizing the UUV when it enters the error circle with a radius of *δ* centered on the UUV’s surface projection, indicating successful tracking. The buoy performs two main tasks in this state. The first task is to synchronize the orientation and position of the UUV. The buoy locks the orientation of the UUV’s water surface projection in real-time when it is within the error circle. If the UUV moves relative to the buoy and the buoy goes out of the error circle, the buoy will track the UUV to compensate for the position deviation, ensuring that the buoy is always within the error circle of the UUV. The second task is to analyze the position coordinates of the UUV. The GNSS positioning system uses the shore base as the reference station, and the buoy as the rover station and the RTK positioning technology is used to achieve high-precision positioning of the buoy [19]. The horizontal distance between the buoy and the UUV is calculated, and it is covered by the NED coordinates relative to the buoy. The latitude and longitude data obtained by the buoy is combined with the NED coordinates of the UUV relative to the buoy to calculate the precise latitude and longitude information of the UUV. The calculation process is as follows:

Step 1. Calculate the horizontal distance between the buoy and the UUV using Equation (11), where Δ*R* represents the horizontal distance between the receiving antenna of the GNSS module on the buoy and the hydrophone A.
(11)$$e\left(t\right)=\Delta R+\sqrt{\rho_{1}^{2}(t)-H^{2}}$$

Step 2. Convert the distance difference into the NED coordinate relative to the buoy using Equations (12) and (13). The NED coordinate system has three parameters: x, y, and z. The parameter x is the due north component of the distance between the target position and the origin of the coordinate system. The parameter y is the due east component of the distance between the target position and the origin of the coordinate system. The parameter z is the vertical downward component of the distance between the target position and the origin of the coordinate system. In this paper, the buoy position and the UUV position are taken as the origin of the coordinate and the target position, respectively. Parameters x, y, and z are replaced with Δ*N*, Δ*E,* and H, respectively. The parameter α represents the yaw of the buoy.
(12)$$\Delta N=e\left(t\right)cos\alpha$$
(13)$$\Delta E=e\left(t\right)sin\alpha$$

Step 3. Fusing the buoy’s latitude and longitude with the NED coordinates of the UUV relative to the buoy. Equations (14) and (15) used for this step take into account the Earth’s variability, denoted by F, as well as the radius of the Earth’s equator $E_{a}$ with a value of 6,378,137 m, the radius of the Earth’s poles $E_{b}$ with a value of 6,356,725 m, and the latitude radius of the buoy’s current position $E_{c}$. In Equations (16)–(18), the altitude, latitude, and longitude of the buoy are represented by $O_{alt}$, $O_{lat}$ and $O_{lon}$, respectively. The altitude, latitude, and longitude of the UUV are represented by $M_{alt}$, $M_{lat}$ and $M_{lon}$, respectively.
(14)$$F=1-\frac{E_{a}}{E_{b}}$$
(15)$$E_{c}=E_{a}(1-F{\times sin}^{2}(O_{lat}))$$
(16)$$M_{alt}=O_{alt}-H$$
(17)$$M_{lat}=\frac{\Delta N}{E_{c}}+O_{lat}$$
(18)$$M_{lon}=\frac{\Delta E}{E_{c}\times cos(O_{lot})}+O_{lon}$$

The geographic coordinates of the UUV are calculated using the fused data and will be transmitted to the ground control station by the buoy to realize UUV positioning.

## 4. Simulation Experiment and Sea Trial Experiment

In order to test the model, a UUV simulation platform was used on an Ubuntu 18.04 system. A gazebo was used to construct an underwater simulation environment, including waves, water flow, and other influencing factors. RVIZ was used to display the coordinates and attitude data. During the simulation process, the rexrov2 (UUV) model was used to simulate UUV, and the desistek_saga model was used to simulate the autonomous tracking buoy. The motion data, tracking, and synchronization performance of the buoy were observed and analyzed. The detailed environment of the simulation experiment is listed in Table 1, and the tracking simulation process is shown in Figure 3.

### 4.1. The Simulation of Capturing UUV

Study the influence of the UUV’s initial horizontal position ${\Delta \theta }_{0}$ relative to the buoy and the capturing accuracy σ of the predicted capture on the capture time. ${\Delta \theta }_{0}$ is in the interval [0°, 175°] with an accuracy of 5°, and the capture accuracy σ is in the interval [0.5,0.95] with an accuracy of 0.05. The experimental results are as follows:

Figure 4 shows the changes of Δ*θ* and the buoy’s yaw over time during the capturing process. The parameter Δ*θ* represents the azimuth of the ROV relative to the buoy. At the beginning, Δ*θ* is 170°, which then drops to approximately 5° after 0.8 s. Capturing is successful when Δ*θ* fluctuates in the range of 10°. Successful capture occurs when Δ*θ* fluctuates within a range of 10°. The capture response is fast. In Figure 5a, *σ* represents the capture accuracy mentioned in Section 3.4 and ${\Delta \theta }_{0}$ represents the initial azimuth of the ROV relative to the buoy at the beginning of capture. It can be observed from Figure 5a that the capture time increases significantly with the increase of ${\Delta \theta }_{0}$ or σ when ${\Delta \theta }_{0}$ is greater than 40°. However, the capture time is always less than 1 s. Figure 5b shows the relationship between the capture error θ and the capture accuracy σ. Generally, as the capture accuracy improves, the capture error significantly decreases. When the capture accuracy is set to 0.5, the mean value of the capture error θ mentioned in Section 3.2 is 48.81°, indicating very poor performance. However, when the capture accuracy is set to 0.95, the mean value of the capture error θ is only 4.72° and the maximum capture error is less than 10°, indicating a great improvement in capturing performance. From the analysis above, it can be concluded that the capture time and capture error have a contradictory relationship with the capture accuracy. The higher the capture accuracy, the longer the capture time and the smaller the capture error, whereas the lower the capture accuracy, the shorter the capture time and the larger the capture error. Since capturing the UUV is the first step towards its positioning, the performance of this step has a critical influence on the process of tracking the UUV and even determines the success of the next step. Therefore, the results and analysis of this experiment have significant reference value for setting an appropriate capture precision to obtain better capture performance. According to the analysis of the results, it is better to set the capture accuracy σ as 0.95 for the next step experiment.

### 4.2. The Simulation of Tracking UUV

The azimuth angle of the UUV relative to the buoy is limited to the capture error range after the buoy successfully captures the target for the first time. The higher the capture accuracy, the smaller the range of the limited azimuth angle of the UUV relative to the buoy, which reflects better tracking performance of the buoy. Therefore, for the convenience of research, the capture accuracy σ is set to 0.95 and the UUV azimuth angle triggering the capture is set to 10°. To investigate the influence of the minimum horizontal azimuth angle (MINHA) of the UUV relative to the buoy when the capture action is terminated, only the movement in the two-dimensional plane in the horizontal direction is studied. The coordinates of the buoy are set to (0,0), and the coordinates of the UUV are set to (20,080), δ is set to 0.5 dm; and the experimental results are as follows:

Figure 6 illustrates the tracking paths of the buoy in a two-dimensional environment under different MINHAs, and Table 2 presents the data on the mean value of the tracking path and the average number of times captured actions occur with different MINHAs. The simulation results clearly show that the buoy can successfully track the static UUV, and capture occurs frequently when the buoy is in close proximity to the UUV. This is because the azimuth angle of the UUV relative to the buoy is more likely to escape from the capture error of the buoy when the buoy is closer to the UUV, thereby triggering the capture of the buoy. The greater the MINHA at the end of the capture action, the smoother the tracking path and the stronger the velocity continuity. This is because a larger MINHA at the end of the capture action reduces the margin for the buoy to capture the UUV’s azimuth, making it easier for the buoy to lose lock to the UUV’s azimuth, which results in less buoy travel between captures. Although the MINHA has little effect on the tracking distance, it will inevitably lead to more captures during the tracking process. To reduce the inertia coefficient of the capture action, it is necessary to appropriately increase the minimum value of the azimuth angle of the UUV relative to the buoy at the end of the capture action and reduce the capturing margin. This would result in stronger velocity continuity and be more suitable for practical applications.

### 4.3. Sea Trial Experiment of Buoy Synchronization Performance

After verifying the buoy’s capture and tracking, the capture accuracy σ value is selected to be 0.95, and the MINHA is set to 3°, 5°, 7°, and 9°. The two devices shown in Figure 7 are used to conduct sea trials in the sea area near the Qinzhou Binhai New Town Wharf to verify the reliability of the autonomous tracking buoy designed in this paper for UUV tracking.

Examining the performance of the buoy’s synchronization of dynamic ROV positions in space. The mean value of the synchronization point position error (MSPPE) refers to the error between the ROV position and the buoy position at the synchronization point, while the mean value of the relative path error (MRPE) refers to the ratio of the difference between the buoy and ROV distance to the ROV distance. These two metrics are used to evaluate the synchronization performance and are compared with the tracking method based on MPC [20]. The experimental results are as follows:

Figure 8a shows the tracking path of the buoy based on ATC-NTP, while Figure 8b shows the trajectory tracking of the ROV based on MPC. The comparison clearly demonstrates that the tracking effect of ATC-NTP is superior to that of MPC. Firstly, the buoy’s trajectory based on ATC-NTP is closer to the ROV’s path than that based on MPC. Secondly, the distance between the buoy and the ROV is much smaller in Figure 8a than that in Figure 8b at the same point in time. indicating that the buoy based on ATC-NTP has better real-time synchronization performance than that based on MPC. The buoy based on ATC-NTP successfully tracks the UUV motion and shows a better tracking effect. A large amount of experimental data is quantified to obtain the index data in Table 3. From the MSPPE index, it can be seen that the positioning error of the buoy relative to the ROV is less than 10 cm and the MRPE is less than 2.5%. These results indicate that the buoy’s path is highly fitted to the ROV motion path, effectively reducing the energy consumption in the tracking process. Thus, the autonomous tracking buoy based on ATC-NTP demonstrates better tracking and real-time synchronization performance.

## 5. Conclusions

This paper discusses the method of dynamic UUV positioning and verifies the feasibility of using autonomous tracking buoys to locate UUV indirectly. The core idea is to precisely track the UUV’s horizontal position using the autonomous tracking buoy. To address the issue of an unknown UUV’s trajectory before positioning, a tracking control algorithm based on ATC-NTP is developed. The simulation and experimental results based on the USBL demonstrate that the proposed method has a better tracking effect on UUVs when applied to buoys with autonomous movement capabilities. The buoy can accurately synchronize the UUV’s horizontal position in space, and the tracking path closely fits the UUV’s motion path with strong real-time performance and low consumption. The UUV’s positioning error is controlled within 10 cm, indicating good positioning performance. PUVV-ATB breaks the premise of trajectory prediction based on traditional tracking control algorithms, providing a new direction for further research on UUV localization. This study’s conclusion has significant potential application value in UUV positioning, underwater synchronous optical communication, and surface tracking of marine organisms. 

## Figures and Tables

**Figure 1 sensors-23-04398-f001:**
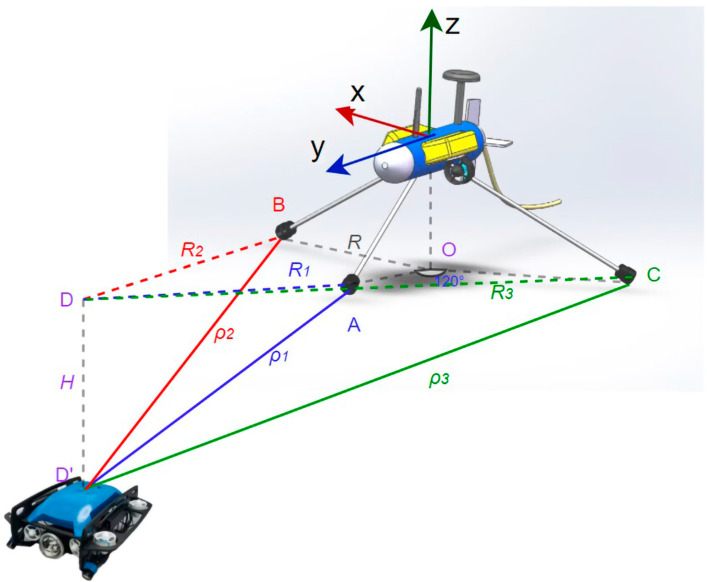
Schematic diagram of the geometric model of buoy and UUV.

**Figure 2 sensors-23-04398-f002:**
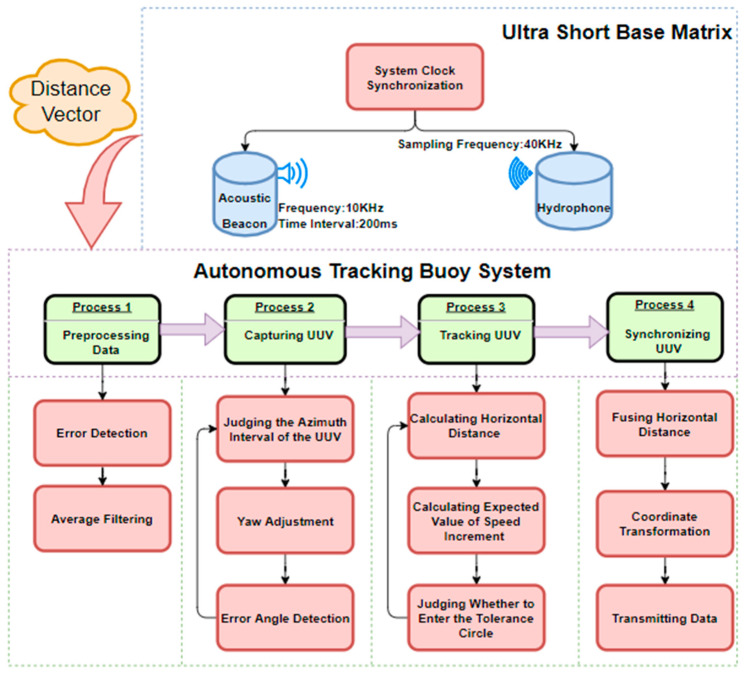
Flow chart of autonomous tracking buoy positioning for UUV.

**Figure 3 sensors-23-04398-f003:**
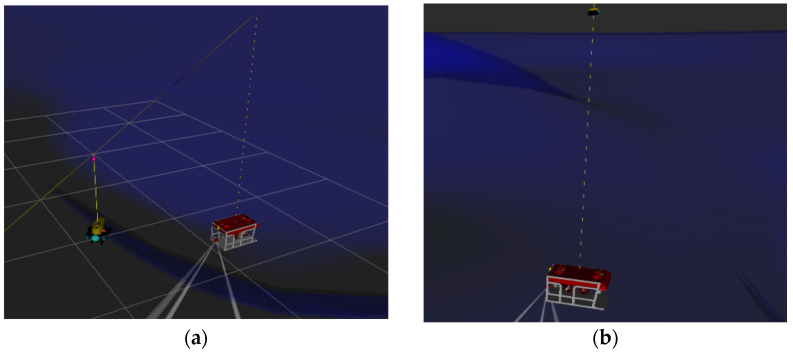
Tracking and synchronization simulation. (**a**) The buoy tracking the UUV; (**b**) the UUV synchronizes with the buoy.

**Figure 4 sensors-23-04398-f004:**
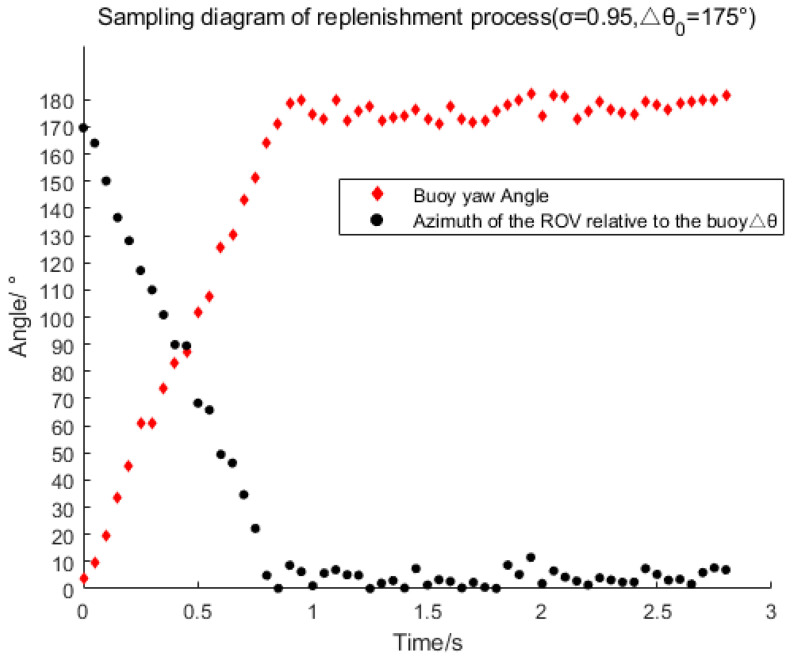
Variation of buoy yaw and azimuth angle Δ*θ* with time during capture.

**Figure 5 sensors-23-04398-f005:**
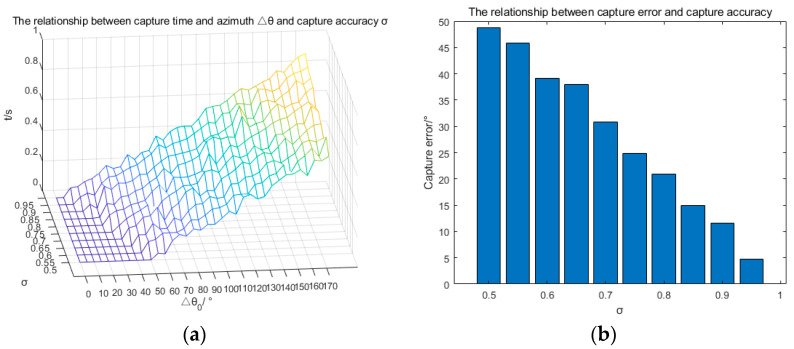
Simulation results of the repeatability experiment. (**a**) The relationship between capture time and ${\Delta \theta }_{0}$ and *σ*; (**b**) the capture error vs. *σ* relationship diagram.

**Figure 6 sensors-23-04398-f006:**
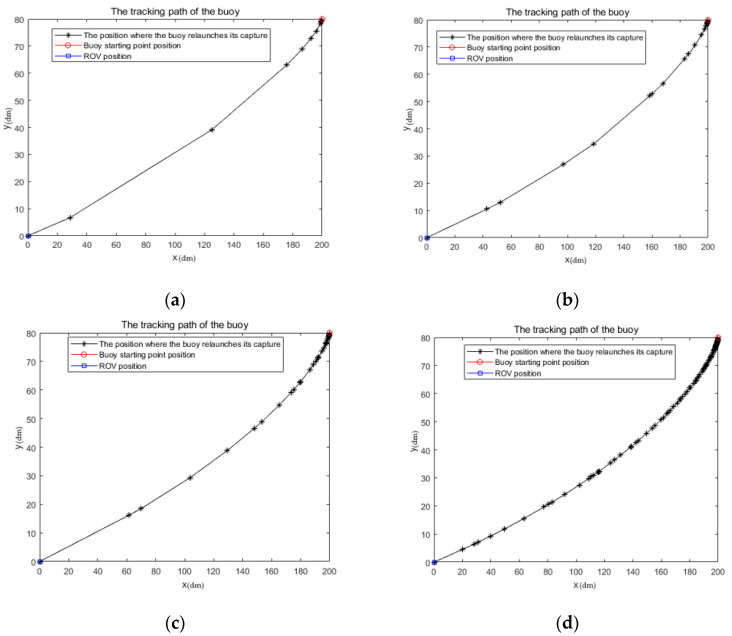
Tracking paths of buoys under different MINHAs. (**a**) MINHA is 3°; (**b**) MINHA is 5°; (**c**) MINHA is 7°; (**d**) MINHA is 9°.

**Figure 7 sensors-23-04398-f007:**
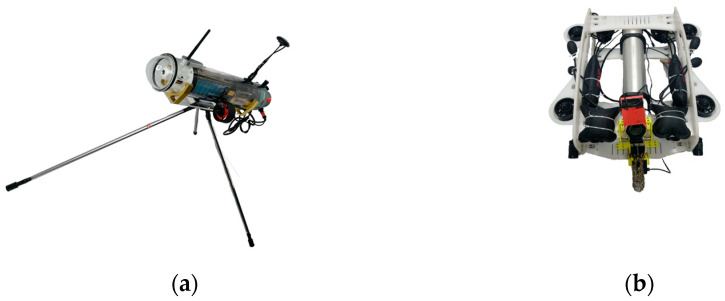
Sea trial experimental equipment. (**a**) Autonomous tracking buoy; (**b**) remoted operated vehicle (ROV).

**Figure 8 sensors-23-04398-f008:**
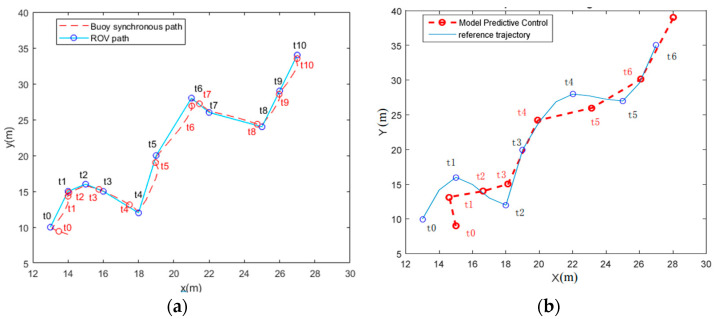
Buoy’s track and ROV’s track. (**a**) ATC-NTP; (**b**) MPC.

**Table 1 sensors-23-04398-t001:** Environment of the simulation experiment.

Computer System	Ubuntu 18.04
CPU	AMD Ryzen 7 PRO 4750U with Radeon Graphics
ROS version	melodic
Gazebo version	9.0.0
RVIZ version	1.13.5
Underwater simulation environment	UUV gazebo world lake

**Table 2 sensors-23-04398-t002:** Metrics of the track performance.

MINHA	Mean Value of Traced Path	Mean Value of Captures
3°	216.5905	13
5°	217.3271	23
7°	217.5285	37
9°	218.0237	119

**Table 3 sensors-23-04398-t003:** Synchronization performance metrics.

MINHA	MSPPE			MRPE
	Distance Error in X Direction (m)	Distance Error in Y Direction (m)	Distance Error of Straight Line (m)	
3°	0.0516	0.0423	0.0667	1.565%
5°	0.0608	0.0493	0.0783	2.074%
7°	0.0634	0.0576	0.0857	2.275%
9°	0.0723	0.0611	0.0947	2.431%

## Data Availability

Not applicable.

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
