# Peer review of "Positioning of Unmanned Underwater Vehicle Based on Autonomous Tracking Buoy"

_sensors, 2023, doi:10.3390/s23094398_

Round 1

Reviewer 1 Report

The article considers the problem of dynamic positioning of UUV using autonomous tracking buoy, which is very important nowadays. Unfortunately, there are some serious flaws in the text, which must be fixed before publication.

1) English language of the text is very poor. Sometimes it is hard to understand the idea of the sentence.

2) Moreover, there seems to be a lack of scientific soundness in this research. For example, authors mention "ultra-short baseline matrix" in the abstract, but do not reveal its meaning and usage mechanism in the text.

I do not think that the article is acceptable for the publication in its current form.

Author Response

Thank you for your valuable comments. Please see the attachment for a point-by-point response to the your comments.

Reviewer 2 Report

This article proposes an adaptive tracking control algorithm without trajectory prediction to improve the positioning of an unmanned underwater vehicle based on an autonomous tracking buoy.

The authors should address the recommendations I have noted in their submitted article, mainly about improving their abstract and introduction section, and add a contribution and survey of the related work sections (Please refer to my comments on the document.)

Author Response

(The authors gave the same response as above.)

Reviewer 3 Report

In order to solve the problem of dynamic positioning of unmanned underwater vehicle with unknown trajectory, the author proposes a method for indirect positioning of unmanned underwater vehicle using autonomous tracking buoy. The manuscript uses tools such as ultra-short baseline matrix, error detection and mean filtering. Avoiding the difficulty of trajectory prediction, and locating unmanned underwater vehicles indirectly.

However, there are still some problems in the manuscript, such as grammar problems, lack of innovation in theoretical formulas, and insufficient detailed analysis of the resulting graph.

In addition, the literature review did not mention the author or unit, therefore, some improvements are needed in the introduction section.

1) In "1. Introduction", the second paragraph of the research introduction on underwater dynamic target tracking, most sentences lack completeness and need a brief description for the author or unit.

2) In Eq. (4) ρij is not explicitly stated in the paper. Generally, graphs and formulas need to be introduced and mentioned in the paper.

3) On line 162&163, the relationship between UUV’s horizontal azimuth deviation angle and σ is not clear enough.

4) In Eq. (11)Eq. (12) △D is not explicitly stated in the paper.

5) On line 241, Step 2. The article lacks an explanation of the relationship between NED coordinates and N、△E.

6) Generally, graphs and formulas need to be introduced and mentioned in the paper. Table.1 and Figure 3 are not mentioned in the text.

7) Section 4.1 The simulation of capturing UUV. The description of the results in this section needs to be more comprehensive and detailed.    Besides, the conclusion drawn from Fig.5 (b) should be written clearly like Fig.5 (a).

8) The name and content of Table 2 are shown in two different pages, hopefully, it could be displayed in one page.

9) This conclusion cannot be clearly seen in Fig.8.

10) There are some minor errors in the format of References, and authors need to modify them according to the requirements of the journal. In addition, there are many cited references that cannot be found.

Author Response

(The authors gave the same response as above.)

Round 2

Reviewer 1 Report

I appreciate the thorough work done by the authors on improving the article. Now it can be accepted for publication in my opinion.

Reviewer 3 Report

This paper has been well revised.